# High Diversity and Low Coinfections of Pathogens in Ticks from Ruminants in Pakistan

**DOI:** 10.3390/microorganisms13061276

**Published:** 2025-05-30

**Authors:** Laila Jamil, Cheng Li, Yifei Wang, Jabran Jamil, Wenya Tian, Di Zhao, Shijing Shen, Yi Sun, Lin Zhao, Wuchun Cao

**Affiliations:** 1Institute of EcoHealth, School of Public Health, Shandong University, Jinan 250012, China; lailashah117@gmail.com (L.J.); licheng97@163.com (C.L.); wangyf419@163.com (Y.W.); 13569319356@163.com (W.T.); zellazdi@163.com (D.Z.); 202321064@mail.sdu.edu.cn (S.S.); 2State Key Laboratory of Pathogen and Biosecurity, Academy of Military Medical Sciences, Beijing 100071, China; 3Livestock and Dairy Development (Ext.) Department, Peshawar 25000, Pakistan; jibran50@gmail.com

**Keywords:** high diversity, phylogenetic analysis, *Anaplasma*, *Ehrlichia*, *Rickettsia*, *Babesia*, *Theileria*, Pakistan

## Abstract

Emerging tick-borne infections pose growing public health threats, causing global disease burdens and economic losses. In this study, tick-borne pathogens were detected in ticks collected from ruminants in 19 sites of Khyber Pakhtunkhwa Province, Pakistan, between 2023 and 2024. A total of 989 ticks, belonging to five species, i.e., *Hyalomma marginatum*, *Rhipicephalus microplus*, *Rhipicephalus sanguineus*, *Rhipicephalus haemaphysaloides*, and *Haemaphysalis bispinosa*, were tested by specific PCR followed by Sanger sequencing. In total, fourteen pathogens including two *Anaplasma* species, three *Ehrlichia* species, three *Rickettsia* species, one *Babesia* species, and five *Theileria* species were identified, with an overall infection rate of 20.2% (95% CI: 17.7–22.7%). Phylogenetic analyses revealed two undefined *Ehrlichia* species: *Candidatus* Ehrlichia hyalommae was exclusively detected in *Hy*. *marginatum* ticks, while *Candidatus* Ehrlichia rhipicephalis was only found in *R. microplus*. Additionally, an undefined *Rickettsia*, provisionally named *Candidatus* Rickettsia pakistanensis, was identified, which is phylogenetically close to *R. sibirica* in North Asia and *R. africae* in Africa, suggesting its potential pathogenicity to humans. Although coinfections of two pathogens were observed, the coinfection rates were quite low. The findings revealed a significant diversity of tick-borne pathogens in Pakistani ticks, which may pose risks to livestock and humans.

## 1. Introduction

Ticks as important vectors transmit a wide range of human and animal pathogens, including bacteria, protozoa, and viruses, and cause substantial disease burden and economic losses worldwide [1]. Emerging tick-borne pathogens are especially a growing global public health threat [2,3]. Pakistan is located in a subtropical zone in South Asia, most parts of which provide favorable environments for ticks, which can infest a variety of hosts and transmit diseases to humans, livestock, and companion animals. In Pakistan, ticks pose a significant challenge for both livestock and human health [4], as they are distributed all over Pakistan’s diverse geographical regions with high abundance, and carry a variety of pathogens [5,6]. Pakistan is home to over 40 distinct species of ticks, with the predominant tick species classified into three genera: *Hyalomma*, *Haemaphysalis*, and *Rhipicephalus* [6]. The high prevalence of *Anaplasma* in buffaloes throughout Khyber Pakhtunkhwa Province constitutes a significant threat to livestock health [7]. The *Hyalomma* and *Rhipicephalus* ticks, particularly *Rhipicephalus microplus*, act as primary vectors for *Anaplasma marginale*, responsible for tick fever in Pakistan [8,9,10]. Similarly, the Crimean–Congo Hemorrhagic Fever Virus (CCHFV) is transmitted by *Hyalomma* ticks and presents a serious risk to public health in Pakistan, resulting in severe hemorrhagic fever, elevated mortality rates, and considerable economic repercussions [11,12,13]. A study conducted from 2017 to 2021 identified *Rickettsia massiliae* in *Rhipicephalus* ticks infesting wild boars in Khyber Pakhtunkhwa Province, indicating the presence of *Rickettsia* in local tick populations, which are associated with adverse effects on animal productivity due to tick-borne diseases [14,15]. In Pakistan, *Theileria annulata* and *Babesia bigemina* exhibit prevalence rates of 16% and 51%, respectively, within cattle populations [9]. Furthermore, research concerning bovine ticks revealed that 31.6% were infected with either *Babesia* or *Theileria* species, with tropical theileriosis, caused by *T. annulata*, being one of the most significant tick-borne diseases affecting bovines [16,17,18,19,20].

Ticks in Pakistan show extensive co-infection in their natural ecosystems. Recent studies indicate that *Hyalomma excavatum*, *Hy*. *isacci*, and *Rhipicephalus decoloratus* exhibit dual-pathogen co-infections, while *Hy*. *scupense* is known to harbor three pathogens. Significantly, four different tick-borne pathogens were identified in ticks from *Hyalomma*, *Haemaphysalis*, and *Rhipicephalus*. These patterns of multi-pathogen presence highlight the complex transmission dynamics within tick-borne disease systems, reinforcing the need for thorough surveillance strategies in endemic areas [6,21].

Khyber Pakhtunkhwa Province has diverse geographical situations, comprising mountains, plains, and valleys. Although the region experiences hot summers and cold winters, the combination of moderate humidity, vegetation cover, and the presence of domestic and wild hosts creates microhabitats that support tick survival. These climatic and geographical characteristics can facilitate the thriving of tick populations and the transmission of tick-borne pathogens [22]. Moreover, livestock production activities, which are the major source of income for rural households and contribute over 60.5% to agricultural value and nearly 11% to the country’s GDP during 2019 and 2020 [23], further offer a favorable ecology for tick infestation. The frequent contact between farmers and livestock certainly increases the transmission opportunities of tick-borne pathogens from animals to humans. However, the situation of ticks and tick-borne infections, as well as their potential risks to public and animal health, has not been well recognized in Khyber Pakhtunkhwa Province, which has hindered the effective control of ticks and tick-borne infections in the region. In this study, we collected ticks from 19 sites of three districts in Khyber Pakhtunkhwa Province, and reported the presence, prevalence, and genetic characteristics of tick-borne agents for establishing target control measures for transmission of potential tick-borne zoonotic pathogens in this region.

## 2. Materials and Methods

### 2.1. Study Areas and Sample Collection

The present study was carried out in Swabi (32.1442° N, 72.4702° E), Swat (35.2220° N, 72.4258° E), and Buner districts (34.4215° N, 72.6150° E) of Khyber Pakhtunkhwa Province in Pakistan in December 2023 to March 2024 (Figure 1). The collection sites’ latitude, longitude, and host animals were recorded during tick collection (Appendix A). Tick collection was carried out with local authorities’ support and approval, including the Civil Veterinary Hospital and Department of Animal Husbandry, Peshawar. Ticks were collected from livestock such as buffaloes, goats, and sheep using approaches that ensured minimal animal stress. These three districts were chosen due to their distinct agroclimatic features. Swabi, at an elevation of approximately 345 m, has a semi-arid subtropical climate with moderate humidity and a dry season extending from November to May. In contrast, Swat and Buner lie above 900 m and experience temperate climates with cooler temperatures and higher rainfall. Livestock in these areas are raised under traditional management systems, including a mix of stall feeding and open grazing, which vary by species and location. These environmental and husbandry differences may influence tick distribution and pathogen transmission.

After immobilizing the animal, the ticks were removed using sterile forceps. The collected tick specimens were then washed with 70% ethanol followed by distilled water to remove contaminants. The ticks were identified by a senior entomologist (Yi Sun) using a stereo microscope (Olympus SZX10, Olympus Corporation, Tokyo, Japan) based on morphological criteria to determine their species and different developmental stages [24,25,26,27]. After identification, the samples were placed in centrifuge tubes, and the ticks were stored at −80 °C before DNA was extracted.

### 2.2. DNA Extraction and PCR Assays

Total DNA of individual ticks was extracted in a separate laboratory using TaKaRa Mini BEST Viral RNA/DNA Extraction Kit Ver. 5.0 (TaKaRa, Dalian, China). Briefly, each tick was placed in a microcentrifuge tube, and buffer VGB, proteinase K, and carrier RNA were added. Then, incubation was performed at 56 °C for cell lysis. Ethanol was added, and the samples were vortexed well and transferred to a centrifuge column, where a series of centrifugation and washing steps were performed in a rotating column according to the manufacturer’s instructions. The DNA was eluted in a final volume of 35 μL of RNase-free water, incubated for one minute at room temperature, and then centrifuged for one minute at 6000× *g* at room temperature. The final step was repeated with the eluent to increase the DNA yield. Extracted DNA was stored at −20 °C before testing.

All tick samples were tested for the presence of *Rickettsia*, *Anaplasma*, *Ehrlichia*, *Borrelia*, *Babesia*, and *Theileria* using specific PCR assays. The specific primers and amplification conditions used for the detection of the various tick-borne agents are shown in Appendix A. Rickettsial species were identified by nested PCR using specific primers to amplify the outer membrane protein A gene (*OmpA*), the *17 kDa* antigen gene (*17 kDa*), and the citrate synthase gene (*gltA*). The genera *Anaplasma* and *Ehrlichia* were detected using semi-nested primers out1/3–17 and out1/out2 targeting the 16S rRNA gene. Positive samples were then detected by PCR targeting citrate synthase (*gltA*) and heat shock protein (*groEL*) gene fragments. PCR targeting the 18S rRNA gene was used to detect *Babesia* and *Theileria*. *Borrelia* were detected by PCR targeting the 5S–23S rRNA intergenic spacer region. All the PCR amplifications were performed using a Veriti 96-Well Thermal Cycler (Applied Biosystems, Waltham, Massachusetts, USA), and all the positive PCR amplicons were confirmed by Sanger sequencing in both directions. Nuclease-free water was used as a negative control in PCR. The obtained nucleotide sequences were then proofread and assembled using CLC Main Workbench 5.0 (Qiagen, Redwood City, CA, USA).

### 2.3. Phylogenetic Analyses

Sequences of each amplified segment by specific PCR was aligned using the BLASTn tool in NCBI (Nucleotide BLAST: Search nucleotide databases using a nucleotide query (nih.gov)) to preliminarily assess the sequences’ similarity and origin to verify the sequencing results’ accuracy. Then, multiple sequence comparisons of the sample sequences with the reference sequences were performed using MAFFT (v7.487) software [28] to ensure the consistency and accuracy of the comparisons. Trial software was used to trim the sequences and remove low-quality regions to ensure the data quality of each fragment [29]. For constructing the joint phylogenetic tree, the results were further pruned using Gblocks comparison to remove unstable sequence regions [30]. The trimmed comparison sequences were subsequently concatenated for subsequent phylogenetic analysis. We used IQ-TREE (v2.1.4) to construct both single-gene and concatenated phylogenetic trees. The software automatically selects the best-fit substitution model and infers phylogenies using the maximum likelihood (ML) method [31]. To assess the confidence level of the phylogenetic tree, an ultra-fast bootstrap method (1000 replications) was used to verify the stability and reliability of the results through multiple random sampling. Finally, the phylogenetic tree was annotated and visualized using the iTOL online tool [32], and genetic distances were calculated by Mega 11 software [33]. The GenBank reference sequences used in the phylogenetic analysis are shown in black in Figures 2–5, and are labeled with their accession numbers, species names, host organisms, and geographic origins.

### 2.4. Statistical Analyses

The positive rate with its 95% confidence interval (95% CI) was estimated for each tick-borne agent in each tick species, and for overall infections of each tick species. Associations between pathogen positivity and tick species were assessed using chi-square (χ^2^) or Fisher’s exact tests where appropriate (i.e., expected frequency < 5). All analyses were performed in R software (v4.4.2), and a two-sided *p*-value < 0.05 was considered statistically significant.

### 2.5. Ethical Approval

The Livestock and Dairy Development Department (ext) in Peshawar, Khyber Pakhtunkhwa, Pakistan approved the ethical protocol for this study (Approval No. 3075). All animal procedures were approved by the relevant ethics committee, ensuring compliance with animal welfare standards.

## 3. Results

### 3.1. Collected Tick Samples

A total of 989 tick species were collected from the 19 sites of three districts in Khyber Pakhtunkhwa Province, Pakistan (Figure 1), comprising 867 females and 122 males from cattle, goats, and sheep (Table 1). The collected ticks belonged to five species of the family Ixodidae, including *Hyalomma marginatum* (157, 15.9%), *Rhipicephalus microplus* (753, 76.1%), *Rhipicephalus sanguineus* (17, 1.7%), *Rhipicephalus haemaphysaloides* (13, 1.3%), and *Haemaphysalis bispinosa* (49, 5.0%). Among them, all the above five tick species were found in Swabi district, four species except *R. sanguineus* were collected from Buner district, and only *R. microplus* was obtained in Swat district.

### 3.2. Detection and Phylogenetic Analysis of Anaplasma

Three gene fragments, 16S rRNA, *gltA*, and *groEL*, were detected by nested PCR assays. Although the positive samples amplified from these three gene fragments were not entirely identical, genetic homology comparison and phylogenetic analysis revealed the presence of two *Anaplasma* species in these tick samples: *Anaplasma marginale* and *Anaplasma ovis* (Figure 2). All the sequences of the PCR products positive for *Anaplasma* were deposited in GenBank, and the accession numbers are listed in Appendix A. The sequences of *A. marginale*-positive samples were closely related to each other, with nucleotide identities of 99.5–100%, 97.2–100%, and 98.2–100% for the 16S rRNA, *gltA*, and *groEL* genes, respectively (Appendix A). *A. marginale* was detected only in *R. microplus* ticks from cattle, in either Swat or Buner districts (Figure 1). In contrast, *A. ovis* was detected in the other four tick species *Hy. marginatum*, *R. sanguineus*, *R. haemaphysaloides*, and *Hae. bispinosa*. All the ticks positive for *A. ovis* were exclusively from sheep, albeit different tick species. These findings reveal a distinct pattern of vector association between the two *Anaplasma* species. *A. marginale* was exclusively associated with *Rhipicephalus microplus*, whereas *A. ovis* was predominantly found in other tick species. Such distribution patterns likely reflect underlying ecological or biological vector–pathogen specificity. This non-random association was strongly supported by statistical analysis (Fisher’s exact test, *p* < 0.001 for both *A. marginale* and *A. ovis*) (Appendix A).

### 3.3. Detection and Phylogenetic Analysis of Ehrlichia

Eleven ticks were positive for both 16S rRNA and *groEL* genes of *Ehrlichia*. All the sequences of amplicons were deposited in GenBank, and the accession numbers are listed in Appendix A. Phylogenetic analyses based on 16S rRNA and *groEL* revealed that a *R. microplus* tick from Swabi was positive for *Ehrlichia minasensis* (Figure 3), the 16S rRNA gene of which shared 99.5% identity with *E. minasensis* strains detected in *R. microplus* from Anhui, China (GenBank accession number: OQ136683.1), and the *groEL* gene fragments were 99.0% similar to those from *R. microplus* from Hubei Province, China (GenBank accession number: OQ511477.1). Three *Hy. marginatum* ticks were positive for an undefined *Ehrlichia* species, the 16S rRNA genes of which were identical with each other and shared 99.4% identity with a previously uncultured *Ehrlichia* species found in Turkey (GenBank accession number: OM065753.1) (Figure 3A). Meanwhile, their *groEL* gene fragments were 99.0–100% identical to each other, and only bore 92.0% similarity to the closest *Ehrlichia* sp. reported in Japan (GenBank accession number: LC385854.1) (Figure 3B). On the phylogenetic tree based on the combined 16S rRNA and *groEL* gene sequences (Figure 3C), the *Ehrlichia* species from the three positive samples formed a distinct branch, suggesting a putative new *Ehrlichia* species. Given that all samples infected with this *Ehrlichia* species were *Hy. marginatum*, it was provisionally named *Candidatus* Ehrlichia hyalommae (Figure 3).

The 16S rRNA gene sequences of the remaining seven *Ehrlichia*-positive samples had 98.9–100% similarity among themselves and 98.3–100% similarity to that of uncultured *Ehrlichia* sp detected in *R. microplus* from Colombia (GenBank accession number: OM475361.1) (Figure 3A). Their *groEL* gene sequences exhibited 99.0–100% similarity to each other and 98.0–99.0% similar to the genetically closest uncultured *Ehrlichia* sp. found in *R. microplus* in China (GenBank accession number: OQ185234.1) (Figure 3B). In the phylogenetic tree based on the combined 16S rRNA and *groEL* gene sequences, the seven positive samples formed a distinctively independent branch (Figure 3C), supporting the notion that it represents an undefined *Ehrlichia* species, which we provisionally designated as *Candidatus* Ehrlichia rhipicephalis because it was found only in *R. microplus* ticks. This *Ehrlichia* species was all detected in *R. microplus* that fed on cattle, although they were distributed in all three districts.

Statistical analyses were conducted to explore potential associations between *Ehrlichia* species and tick vectors. Although *E. minasensis* and *Candidatus* E. rhipicephalis were detected exclusively in *R. microplus*, these associations were not statistically significant (Fisher’s exact test, *p* = 1 and *p* = 0.783, respectively) (Appendix A), possibly due to the limited number of positive samples. In contrast, *Candidatus* E. hyalommae exhibited a statistically significant association with *Hy. marginatum* (*p* = 0.022), suggesting a potential vector preference. These findings indicate species-specific trends in vector association among *Ehrlichia* spp., although larger sample sizes are needed to confirm these patterns with greater confidence.

### 3.4. Detection and Phylogenetic Analysis of Rickettsia

To screen for *Rickettsia* infections in ticks, we amplified fragments of the *OmpA*, *17 kDa*, and *gltA* genes using nested PCR and detected six samples positive for all three genes and eight samples positive for two of three genes. All the sequences of the detected *Rickettsia* segments were deposited in GenBank, and the accession numbers are listed in Appendix A. Although the positive samples amplified from the three genes fragments were not entirely identical, genetic homology comparison and phylogenetic trees constructed based on *OmpA*, *17 kDa*, and *gltA* genes identified three *Rickettsia* species. Among them, *Rickettsia felis* was identified in an *R. microplus* tick, showing 99.8% identity to previously reported *R. felis* sequences from *Ctenocephalides felis* in Guatemala for the *OmpA* gene (GenBank accession number: JN990593.1) and 100% identity to *R. felis* sequences from *Ctenophthalmus agyrtes* in Lithuania for the *17 kDa* gene (GenBank accession number: MF491767.1) (Figure 4A,B). Phylogenetic trees constructed based on the *OmpA*, *17 kDa*, and *gltA* gene fragments revealed that the *Rickettsia* detected in *R. sanguineus* and *R. microplus* ticks clustered within the same branch as *Rickettsia massiliae*, indicating that they belong to the same species (Figure 4). Specifically, *R. sanguineus* ticks infected with *R. massiliae* were all from goats or sheep in Swabi district, while the positive *R. microplus* tick was from cattle in Buner district.

Another *Rickettsia* exhibited a similarity of 100%, 99.2–100%, and 98.3–100% similarity in *OmpA*, *17 kDa*, and *gltA* genes sequences, respectively. In the phylogenetic trees based on *OmpA* and *gltA* genes, they were clustered with *Rickettsia africae*, with similarities of 99.6–99.8% and 97.4–100%, respectively (Figure 4A,C). However, in the *17 kDa* phylogenetic tree, they were in the same branch as known human-derived *Rickettsia sibirica* strains, with 100% identity (Figure 4B). To further validate the evolutionary position of the *Rickettsia*, a phylogenetic tree on the basis of combined *OmpA*, *17 kDa*, and *gltA* gene sequences was constructed, and revealed that the *Rickettsia* from positive samples in this study formed a separate branch (Figure 4D), distinctive from *R. sibirica* and *R. africae*, suggesting that they represent a new *Rickettsia* species. Here, we provisionally named it *Candidatus* Rickettsia pakistanensis, in honor of discovering this new *Rickettsia* species in Pakistan. Notably, *Candidatus* Rickettsia pakistanensis was only detected in *Hy. marginatum* ticks parasitizing goats or sheep in the Swabi district.

### 3.5. Detection and Phylogenetic Analysis of Babesia and Theilria

To screen for *Babesia* and *Theileria* infections in ticks, we amplified the 18S rRNA gene using a pair of universal primers, and proved the test results through Sanger sequencing of PCR products (Appendix A). Based on similarity comparison and phylogenetic analysis of 18S rRNA gene sequences, one species of *Babesia* and five species of *Theileria* were identified, namely *Babesia bigemina*, *Theileria annulata*, *Theileria buffeli*, *Theileria luwenshuni*, *Theileria orientalis*, and *Theileria sinensis* (Figure 5). The two samples positive for *B. bigemina* were detected in *R. microplus* ticks from cattle in Swat district. The 18S rRNA gene of the 43 *T. annulata* sequences had 98.3–100% nucleotide identity with previously reported *T. annulata* sequence from Iran (GenBank accession number: HM628581.1), clustering into one branch on the phylogenetic tree. All 43 *T. annulata* detected in this study were in *R. microplus* ticks from cattle and sheep in all three districts. The 24 *T. orientalis* sequences shared 99.2–100% identity with each other and with known *T. orientalis* sequence in Myanmar (GenBank accession number: MG599099.1). Among the 24 *T. orientalis*-positive ticks, 22 were *R. microplus* from the three districts, while the other 2 positive samples included *Hae. bispinosa* and *Hy. marginatum* from Swabi district. There were 14 sequences belonging to *T. luwenshuni*, which shared 98.3–100% identity with each other and 98.4–100% identity with *T. luwenshuni* reported in India (GenBank accession number: MG194417.1). The tick species for *T. luwenshuni* detected in the Swabi district were *Hy. marginatum* (seven samples) and *Hae. bispinosa* (five samples), with goats (or sheep) as the hosts. *T. luwenshuni*-infected *R. microplus* ticks were all from the Buner district, indicating that *T. luwenshuni* infections might be distributed in different tick species of different regions. Twelve samples were detected with *T. sinensis*, sharing 96.9–100% identity with previously reported *T. sinensis* sequences found in *Ixodes persulcatus* (GenBank accession number: OR244360.1) from Pakistan. All *T. sinensis*-positive samples were from *Hae. bispinosa* in the Swabi district. Only one sample was detected with *T. buffeli*, sharing 100% similarity with the *T. buffeli* detected in Buffalo from India (GenBank accession number: OR067901.1).

### 3.6. Prevalence and Co-Infections of Tick-Borne Agents

As mentioned above, this study detected a total of 14 agents in 989 ticks across the three districts. For *Anaplasma*, *Ehrlichia*, and *Rickettsia*, two to three gene fragments were tested. To more accurately determine the infection rate of these tick-borne agents, a sample was considered positive only if at least two gene fragments of the same agent were detected. As a result, 200 ticks tested positive for at least one agent, with an overall positive rate of 20.2% (95% CI: 17.7–22.7%). The infection rates of *Hy. marginatum*, *R. microplus*, *R. sanguineus*, *R. haemaphysaloides*, and *Hae. bispinosa* were 40.1% (95% CI: 32.5–47.8%), 17.3% (95% CI: 14.6–20.0%), 47.1% (95% CI: 23.3–70.8%), 7.7% (95% CI: 3.5–11.8%), and 40.8% (95% CI: 27.1–54.6%), respectively (Table 2). *R. microplus* carried nine agents, followed by *Hy.marginatum* (seven) and *Hae. bispinosa* (four). *T. orientalis* and *T. luwenshuni* were detected in three tick species, and *R. massiliae* and *T. sinensis* in two tick species, respectively. The other nine tick-borne agents were found each in one tick species (Table 2 and Table 3).

A total of 22 out of 989 ticks were positive for two agents (Table 4), with an overall co-infection rate of 2.2% (95% CI: 1.31–3.14%). None of the ticks was infected by three or more agents. The co-infection of *A. marginale* and *T. annulata* in *R. microplus* was most commonly seen, but only observed in 1.5% ticks. Notably, the occurrence of co-infections was associated with specific tick species and districts. Co-infections of *A. marginale* with *B. bigemina* and *A. marginale* with *T. annulata* were only found in *R. microplus* parasitizing cattle in the Swat district. Co-infections of *A. ovis* with *Candidatus* Rickettsia pakistanensis were detected in *Hy. marginatum*, and co-infections of *A. ovis* and *T. sinensis* occurred in both *Hy. marginatum* and *Hae. bispinosa* in Swabi district (Table 4). To assess whether the co-infection of *A. marginale* and *T. annulata* in *R. microplus* occurred more frequently than expected by chance, we conducted a 2 × 2 Chi-square test. The result revealed a statistically significant association between the two pathogens (χ^2^ = 22.49, df = 1, *p* = 2.11 × 10⁻⁶), indicating that this co-infection occurred more often than expected under random distribution. The contingency table and expected frequencies are provided in Appendix A.

## 4. Discussion

In this study, *Rhipicephalus* ticks were observed as the most frequently occurring, followed by the *Hyalomma* and *Haemaphysalis* ticks. This pattern of tick infestation aligns with prior studies conducted in different agro-ecological regions areas in Khyber Pakhtunkhwa Province, Pakistan [34]. Pakistan’s hot and humid climate might be conducive to the proliferation of ticks and their associated pathogens [8,35]. A total of 14 agents were identified in parasitic ticks on livestock, including 3 *Rickettsia* species, 2 *Anaplasmsa* species, 3 *Ehrlichia* species, 1 *Babesia* species, and 5 *Theilria* species, with an overall positive rate of 20.2%. Except the 3 undefined tick-borne agents, the other 11 are known pathogens of humans and animals. Notably, the phylogenetic association of *Candidatus* Rickettsia pakistanensis with *R. sibirica*, known to cause tick-borne typhus in North Asia, and *R. africae*, responsible for tick bite fever in Africa, indicates a potential zoonotic risk. The phylogenetic trees based on 16S rRNA and *groEL* genes revealed two undefined *Ehrlichia* species, *Candidatus* E. hyalommae and *Candidatus* E. rhipicephalis, each of which was situated in a distinct branch, and exclusively identified in *Hy. marginatum* and *R. microplus*, respectively. Their pathogenicity to animals and humans requires further investigation. The high diversity in tick-borne bacteria and protozoan underscores the intricate ecological network of tick-borne pathogens in the region, which can present a dual threat to both the livestock economy and human health. The proactive surveillance of ticks and tick-borne pathogens should be enhanced in the region to alleviate the risk of tick-borne diseases in humans and livestock.

The tick-borne pathogens seem to be specific to tick species. Our findings reveal species-specific associations between tick-borne pathogens and their tick vectors, suggesting ecological or biological specificity in these vector–pathogen systems. *A. ovis* has a broad range of tick vectors; four (*Hy. marginatum, Hae. bispinosa, R. sanguineus*, and *R. haemaphysaloides*) of five tick species in this study are positive for the pathogen, which causes diseases in sheep and goats. This pattern highlights the potential for *A. ovis* to circulate among diverse tick species infesting small ruminants. In contrast, *A. marginale* is detected exclusively in *R. microplus*. This strong and statistically significant association (Fisher’s exact test, *p* < 0.001) suggests a degree of vector–pathogen adaptation. *R. microplus* is an important vector of *A. marginale* in several tropical and subtropical countries and other regions of Pakistan [36], possibly due to the pathogen’s adaptation to host immune tolerance. Similarly, our analysis of *Ehrlichia* species also indicated possible species-specific vector preferences. *E. minasensis*, a widely distributed pathogen known to cause clinical ehrlichiosis in dogs [37], is detected exclusively in *R. microplus*, suggesting a potential association, although not statistically significant (*p* = 1), possibly due to limited sample size. Two novel *Ehrlichia* are identified in this study, designated as *Candidatus* E. hyalommae and *Candidatus* E. rhipicephalis. *Candidatus* E. hyalommae is exclusively identified in *Hy. marginatum* of Swabi district, and this association was statistically significant (*p* = 0.022), suggesting a potential ecological specialization. In contrast, *Candidatus* E. rhipicephalis is identified across all three districts, but only in *R. microplus* ticks, yet the association was not statistically significant (*p* = 0.783). These findings highlight the possible vector-specificity of certain *Ehrlichia* species and underscore the need to broaden the detection range and conduct further molecular studies to elucidate the genetic characteristics, distribution, and epidemiological significance of the two novel species.

*R. massiliae* is detected in both *R. microplus* and *R. sanguineus* ticks, and it can cause a disease known as Mediterranean spotted fever [38,39], emphasizing the need for enhanced surveillance among human population. The newly identified “*Candidatus* R. pakistanensis” is only detected it in *Hy. marginatum* infesting goats (or sheep) in Swabi district, and is phylogenetically related to human-pathogenic *R. africae* (causing African tick bite fever) and *R. sibirica* (causing Siberian tick typhus) [38], suggesting its potential risk for infecting humans. It is important to conduct additional surveillance studies to investigate their animal hosts and geographic range, as well as their potential zoonotic implications. One *R. microplus* from Swabi tested positive for *E. minasensis*, which has been detected in *R. sanguineus* and *R. microplus* ticks infesting stray dogs in Pakistan [40], indicating it has a wider range of vectors and animal hosts.

This study identifies one species of *Babesia* (*B. bigemina*) and five species of *Theileria*, illuminating the diverse range of tick-borne protozoa in this province. Although evidence of *B. bigemina* in cattle has been reported in Pakistan [41], this study directly confirmed its presence in tick vectors through molecular detection. Among detected *Theileria* species, *T. annulata* exhibits the highest infection rate, consistent with findings from livestock-dense areas such as Punjab, Pakistan, further confirming its role as the primary causative agent of tropical theileriosis in cattle [42,43]. Notably, the detection of *T. sinensis* addresses the first detection of the Asian-specific *Theileria* specie in Pakistan, emphasizing the necessity to reassess the distribution patterns of *Theileria* throughout the country. Additionally, the detection of *T. luwenshuni* holds particular significance, as this species is traditionally associated with sheep and goat infections [44], suggesting potential cross-species transmission risks in the region, likely linked to mixed grazing practices in agro-pastoral transition zones. This study reveals the diversity of tick-borne protozoa in Khyber Pakhtunkhwa Province; however, comparisons with other areas of Pakistan indicates significant geographical differences. For instance, the reported infection rate of *T. annulata* in Sindh Province reached 12–15% [43], significantly higher than the 4.3% observed in this study. This discrepancy may be attributed to variations in humidity, livestock density, and tick control measures. Furthermore, the low prevalence of *T. buffeli* (0.10%) stands in contrast to studies conducted in the Middle East [45], indicating the possibility of distinct limiting factors influencing ecological adaptation or host specificity of the pathogen within the context of Pakistan.

In spite of the high diversity of tick-borne agents in this study, their co-infection rates are quite low, with no co-infection of three or more agents observed. The presence of co-infection is associated with specific tick species and districts. The co-infections of *Anaplasma* species (*A. marginale* and *A. ovis*) and either a protozoa or *Candidatus* R. pakistanensis are present in *R. microplus*, *Hy. Marginatum*, and *Hae. bispinosa* ticks (Table 4), suggesting potential interactions between these pathogens within the three tick species. These results are similar to two previous studies in China and Thailand, where co-infection with *Anaplasma* and *Rickettsia* were observed in *Dermacentor* and *Amblyomma* ticks [46,47]. In particular, persistent infection of *A. ovis* is associated cyclical fluctuations in Rickettsial levels, which can alter the infection rate of ticks and thus influence transmission [48]. The low overall co-infection rate may reflect ecological segregation or differing vector and host preferences among pathogens. However, one notable exception is the statistically significant co-infection pattern observed between *A. marginale* and *T*. *annulata* in *R. microplus*. This non-random association suggests that these two pathogens may share similar ecological niches or transmission routes. The co-infection may arise from simultaneous exposure to infected cattle, or potentially through facilitative mechanisms within the tick that enhance dual colonization. While the precise biological interactions remain to be elucidated, such co-infections may affect pathogen persistence, host pathogenicity, and vector competence. These findings highlight the complexity of tick–pathogen interactions and underscore the importance of considering co-infection dynamics in surveillance and control programs. Further studies are needed to investigate the underlying mechanisms of pathogen–pathogen interactions and their epidemiological consequences.

The study has several limitations. It was geographically limited to three districts in Khyber Pakhtunkhwa Province, Pakistan, potentially missing pathogen diversity in other regions. Tick samples were collected only from livestock, excluding potential pathogens in wildlife or other animal species. The short sampling period did not account for seasonal variations in tick populations and pathogen prevalence. Additionally, PCR-based detection may have missed some pathogens, especially in coinfected ticks, leading to an underestimation of pathogen diversity.

In Pakistan, there remain major knowledge gaps regarding various ticks. Therefore, pathogen detection and genetic diversity analysis for ticks parasitizing ruminants were conducted in three districts of Khyber Pakhtunkhwa Province, Pakistan. Our findings reveal the identification of 14 tick-borne pathogens from five tick species in three genera in Pakistan, including *Anaplasma*, *Ehrlichia*, *Rickettsia*, *Babesia*, and *Theileria*. Notably, in addition to known pathogens, a novel *Rickettsia* species and two novel *Ehrlichia* species were detected in this study. These findings highlight the high diversity and complexity of tick-borne pathogens in the region, while differences in host adaptation and regional distribution between different pathogen species highlight the complexity of tick-borne pathogen transmission. The results of this study contribute to understanding the characteristics, molecular epidemiology, and geographical distribution of ticks and their associated pathogens in Pakistan and are important for developing targeted and effective control measures to control tick-borne diseases in livestock and humans.

## 5. Conclusions

This study provides critical insights into the diversity, host specificity, and geographic distribution of tick-borne pathogens in Khyber Pakhtunkhwa Province, Pakistan. A total of 14 pathogens—including bacteria and protozoa—were detected across five tick species, with notable discoveries including one novel *Rickettsia* and two novel *Ehrlichia* species. The predominance of *Rhipicephalus* ticks and the detection of zoonotic agents such as *Candidatus* Rickettsia pakistanensis underscore the potential public health implications. The species-specific associations, limited co-infection rates, and regional variations observed in this study highlight the complexity of tick–pathogen interactions and the influence of ecological factors. These findings emphasize the urgent need for enhanced, region-wide surveillance and targeted control strategies to mitigate the risk of tick-borne diseases in both livestock and human populations.

## Figures and Tables

**Figure 2 microorganisms-13-01276-f002:**
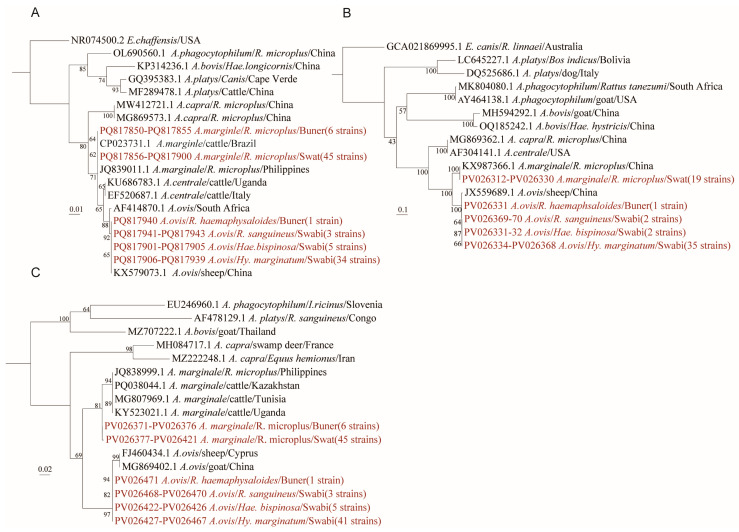
Phylogenetic trees of *Anaplasma* were constructed using the maximum likelihood (ML) method with 1000 bootstrap replicates based on (**A**) 16S rRNA, (**B**) *gltA*, and (**C**) *groEL* genes. The evolutionary positions of samples that tested positive for at least two of the three gene fragments are highlighted in red text.

**Figure 3 microorganisms-13-01276-f003:**
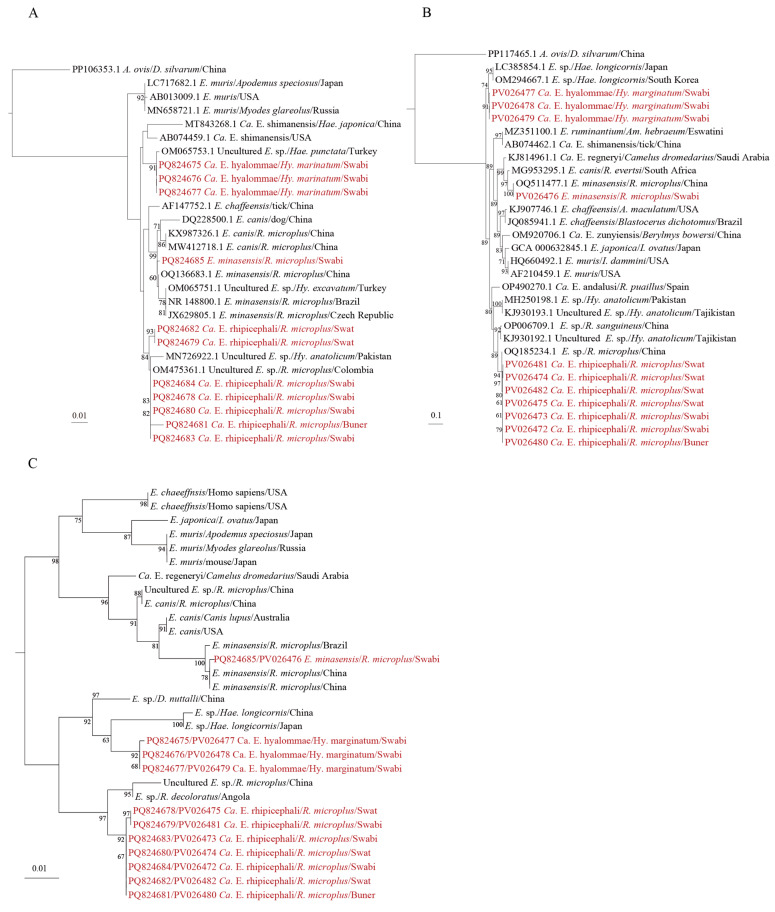
Phylogenetic trees of *Ehrlichia* were constructed using the maximum likelihood (ML) method with 1000 bootstrap replicates. Separate trees were generated for (**A**) 16S rRNA and (**B**) *groEL* genes. (**C**) A concatenated tree was built based on 16S rRNA and *groEL* nucleotide sequences. The evolutionary positions of samples that tested positive for all two gene fragments are highlighted in red text.

**Figure 4 microorganisms-13-01276-f004:**
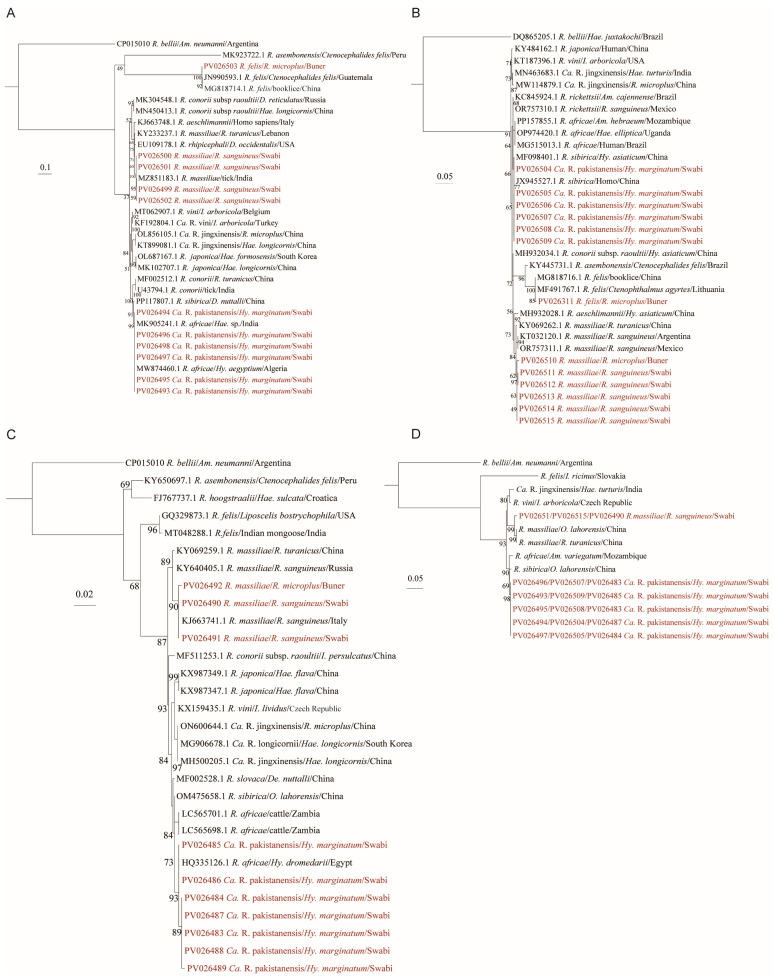
Phylogenetic trees of *Rickettsia* were constructed using the maximum likelihood (ML) method with 1000 bootstrap replicates. Separate trees were generated for (**A**) *OmpA*, (**B**) *17 kDa*, and (**C**) *gltA* genes. (**D**) A concatenated tree was built based on *OmpA*, *17 kDa*, and *gltA* nucleotide sequences. The evolutionary positions of samples that tested positive for at least two of the three gene fragments are highlighted in red text.

**Figure 5 microorganisms-13-01276-f005:**
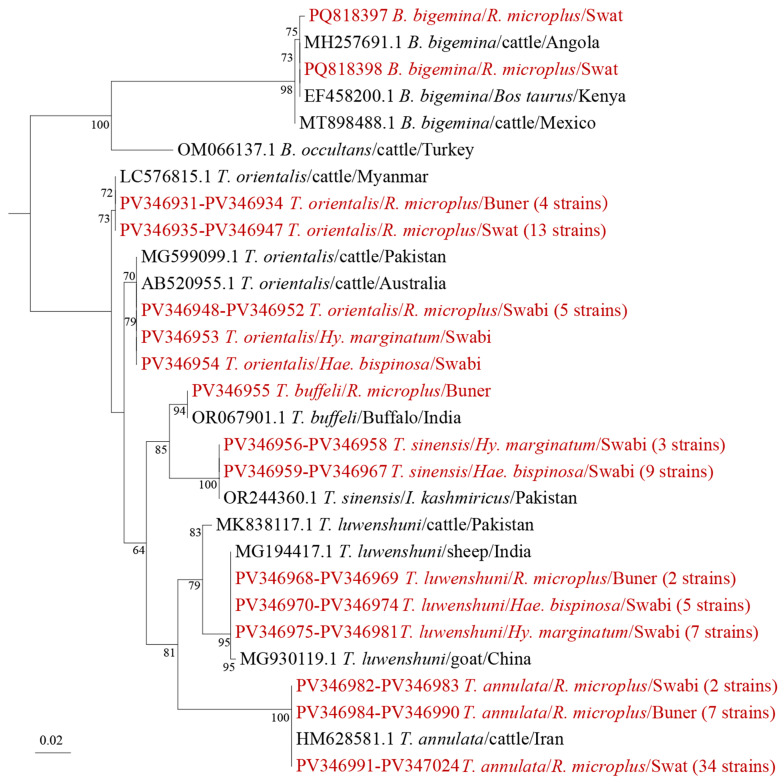
Phylogenetic trees of *Theileria* and *Babesia* strains based on their 18S rRNA gene sequences. Sequences obtained in this study are highlighted in red text.

**Figure 1 microorganisms-13-01276-f001:**
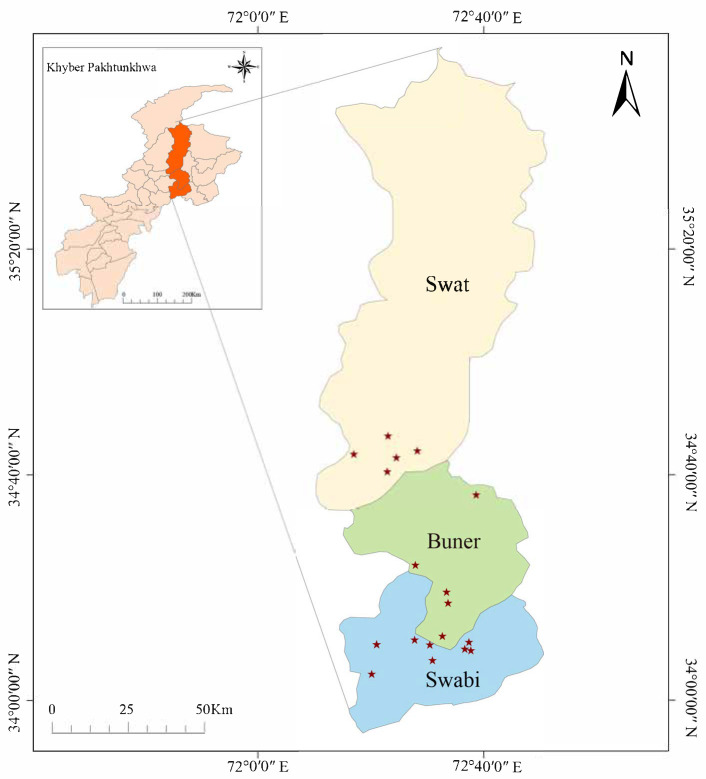
Distribution of tick samples in Khyber Pakhtunkhwa Province, Pakistan. The red stars indicated in the diagram designate 19 specific sampling sites located across three districts within the Khyber Pakhtunkhwa Province of Pakistan.

**Table 1 microorganisms-13-01276-t001:** Information on tick samples collected from different hosts in three districts of Khyber Pakhtunkhwa Province, Pakistan.

District	Host	Tick Species					Total
		*Rhipicephalus*			*Haemaphysalis*	*Hyalomma*	
		*Rhipicephalus* *microplus*	*Rhipicephalus sanguineus*	*Rhipicephalus haemaphysaloides*	*Haemaphysalis bispinosa*	*Hyalomma* *marginatum*	
Swabi	Cattle	156 (F122, M34) *	-	-	3 (M3)	155 (F105, M50)	314
	Goat/Sheep	-	17 (F9, M8)	9 (F9)	45 (F40, M5)	-	71
Buner	Cattle	114 (F114)	-	-	-	-	114
	Sheep	101 (F95, M6)	-	4 (M4)	1 (M1)	2 (F2)	108
Swat	Cattle	382 (F371, M11)	-	-	-	-	382
Total	Sheep/Goat/Cattle	753	17	13	49	157	989

* F means female; M means male.

**Table 2 microorganisms-13-01276-t002:** Positive rate of tick-borne bacteria in ticks from ruminants, Pakistan.

Tick Species	No. of Tested Ticks	Positive Rate % (95% CI)							
		*Anaplasma*		*Ehrlichia*			*Rickettsia*		
		*A.* *marginale*	*A.* *ovis*	*E.* *minasensis*	*C.* E. hyalommae	*C.* E. rhipicephalis	*R.* *felis*	*R.* *massiliae*	*C.* R. pakistanensis
*Hyalomma marginatum*	157	0	26.1 (19.2–32.9)	0.64 (0.02–3.5)	1.92 (0.4–5.5)	0	0	0	4.5 (1.2–7.7)
*Rhipicephalus microplus*	753	6.8 (5.0–8.6)	0	0	0	0.93 (0.2–1.6)	0.1 (0.0–0.7)	0.1 (0.0–0.7)	0
*Rhipicephalus sanguineus*	17	0	17.6 (4.1–41.9)	0	0	0	0	29.4 (10.3–55.8)	0
*Rhipicephalus haemaphysaloides*	13	0	7.7 (0.2–35.8)	0	0	0	0	0	0
*Haemaphysalis bispinosa*	49	0	10.2 (3.4–22.2)	0	0	0	0	0	0
Total	989	5.2 (3.7–6.5)	5.1 (3.7–6.4)	0.1 (0.0–0.6)	0.3 (0.1–0.8)	0.7 (0.2–1.2)	0.1 (0.0–0.6)	0.6 (0.1–1.1)	0.7 (0.2–1.2)

**Table 3 microorganisms-13-01276-t003:** Positive rate of tick-borne protozoans in ticks from ruminants, Pakistan.

Tick Species	No. of Tested Ticks	Positive Rate % (95% CI)					
		*Babesia*	*Theileria*				
		*B.* *bigemina*	*T.* *annulata*	*T.* *buffeli*	*T.* *luwenshuni*	*T.* *orientalis*	*T.* *sinensis*
*Hyalomma marginatum*	157	0	0	0	4.5 (1.2–7.7)	0.64 (0.02–3.5)	1.92 (0.4–5.5)
*Rhipicephalus microplus*	753	0.3 (0.0–0.9)	5.7 (4.0–7.4)	0.1 (0.0–0.7)	0.3 (0.0–0.9)	3.0 (1.7–4.1)	0
*Rhipicephalus sanguineus*	17	0	0	0	0	0	0
*Rhipicephalus haemaphysaloides*	13	0	0	0	0	0	0
*Haemaphysalis bispinosa*	49	0	0	0	10.2 (3.4–22.2)	2.0 (0.0–11.0)	18.8 (7.6–29.2)
Total	989	0.2 (0.0–0.5)	4.3 (3.0–5.6)	0.1 (0.0–0.6)	1.4 (0.7–2.2)	2.4 (1.5–3.4)	1.2 (0.5–1.9)

**Table 4 microorganisms-13-01276-t004:** Co-infect rate of pathogens in ticks from ruminants, Pakistan.

Co-Infection Pathogens	Tick Species	Host	District	Number	Positive Rate % (95% CI)
*A. marginale* + *B. bigemina*	*R. microplus*	Cattle	Swat	2	0.2 (0.0–0.5)
*A. marginale* + *T. annulata*	*R. microplus*	Cattle	Swat	11	1.1 (0.5–1.7)
*A. ovis* + *C.* R. pakistanensis	*Hy. marginatum*	Goat/Sheep	Swabi	1	0.1 (0.0–0.6)
*A. ovis* + *T. luwenshuni*	*Hy. marginatum*	Goat/Sheep	Swabi	1	0.1 (0.0–0.6)
*A. ovis* + *T. sinensis*	*Hae. bispinosa*	Goat/Sheep	Swabi	3	0.3 (0.1–0.8)
	*Hy. marginatum*	Goat/Sheep	Swabi	3	0.3 (0.1–0.8)
*C.* R. pakistanensis + *T. luwenshuni*	*Hy. marginatum*	Goat/Sheep	Swabi	1	0.1 (0.0–0.6)

## Data Availability

The datasets supporting the conclusions of this article are included within the article and Appendix A.

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
