# Peer review of "High Diversity and Low Coinfections of Pathogens in Ticks from Ruminants in Pakistan"

_microorganisms, 2025, doi:10.3390/microorganisms13061276_

Round 1
Reviewer 1 Report
Comments and Suggestions for Authors
Comments to the paper” High diversity and low coinfections of pathogens in ticks from ruminants, Pakistan”
General comments
The paper describes the outcome of a survey on tick and tick-borne bacteria and protozoans. The approach could be characterized as a simple bulk approach as the authors simply harvested a considerable number of ticks from infested ruminants, subjected them to microbial analysis, and reported the prevalence. It is therefore methodologically and statistically a relatively uncomplicated paper.
Title: make it into a sentence…. of pathogens in ticks from ruminants in Pakistan
Authors
The list of authors includes 10 named people but finishes off with an “and” suggesting that one is missing.
There is a long list of email addresses in the affiliation which seems out of place.
Introduction
The introduction is relatively short and does not provide state-of-the-art for tick research in Pakistan. With just seven references, the reader is left ignorant of the fact that hundreds of papers have been published in recent years. A search in Google Scholar using the terms “ticks khyber pakhtunkhwa Pakistan” return more than 2000 related papers. It is impossible to determine whether the authors observed a high or low diversity (as suggested in the title) when there is no comparison with other studies in Pakistan or elsewhere.
Materials and methods
The materials and method section are fine regarding the chemical analysis and sequencing of the DNA. It would, however, be desirable if the authors added information on the habitats, their altitude, and meteorological information, just as a brief description of the pastures would be welcomed. Surely there would be differences between the pastures used for sheep/goats and cattle, which might explain their findings.
Results
The results are clearly and systematically explained in the text. Sadly, the figures are in such small print that they are unreadable, and Table 1 is quite a challenge because the decimals shift between lines. Pls, enlarge the figures and split the table in two – one for bacteria and one for protozoans.
The authors state that the number of coinfections is low, but do not provide evidence to the fact. Pls., add a 2x2 frequency table (Chi2-test) showing that the cooccurrence of A. marginale and T.annulata in R. microplus - is below the expected – and if it is not significant remove the comments.
Discussion
As for the introduction - it seems a bit short and the references are few.
Specific comments.
Line 1. It's difficult to say whether the diversity is high because you have not made any comparison, No tests show that the number of coinfection is fewer than expected from the prevalence rates.
Line 4. And “who”
Line 6: delete email addresses.
Line 14: delete “Ruminants are key in pathogen transmission”
Line 36: Should be “Pakistan is located in…”
Line 44: Ticks thrive almost everywhere and “hot summers and cold winters”, do not offer an ideal environment for tick survival. Be more specific.
Line 53: a hindrance ??
Line 66. Insert habitat descriptions and meteorology
Line 72: write ⁰C – do not spell out.
Line 73: delete “separately”
Line 115: name the program that was used.
Line 125. Move Table S2 from the supplemental material and place it on this page.
Line 130: The text in Figure 1 is too small – unreadable.
Line 130: The text in Figure 2 is too small – unreadable.
Line 168: The text in Figure 3 is too small – unreadable.
Line 204: The text in Figure 4 is too small – unreadable.
Line 216: Phylogenetic – not evolutionary
Line 225: babesia bigemina – loose the “r” – also elsewhere
Line 268: Split the table in two – Table 1 and Table 2, such that the numbers are readable
Line 275: Table 2 is split over two pages – fix it.
Line 278: in the text associated with Table 2, add a 2x2 frequency table (Chi2-test) showing that the cooccurrence of A. marginale and T.annulata in R. microplus - is below the expected.
Line 294: Requires – not required
Line 300. It would be nice with a statistical analysis that supports this proposition.
Line 320: very few things in life are imperative – so a lesser verbal statement will do.
Line 326: I never heard of “Intricate diversity”.
Line 341: Rephrase: “climate humidity”
Line 356: to suggest competitive dynamics for species of bacteria that are swimming in resources seems quite odd. Pls, note that positive and negative associations typically occur simply because organisms like or dislike the given conditions.
Line 365: Perhaps you could mention viruses in the list of missing items.
Reviewer 2 Report
Comments and Suggestions for Authors
The manuscript “High diversity and low coinfections of pathogens in ticks from ruminants, Pakistan” is well written and follows a logical sequence; however, some text errors and scientific names are misspelled (for example, Theilria or Babesia birgemina). The content presents essential information on the prevalence of different genera causative agents of tick-borne diseases by ticks affecting ruminants in Pakistan. However, I believe that improvements can be made before publication.
Line 65 and 123-124: Are all domestic ruminant species included? I suggest listing the animal species from which the study collected ticks. Failure to provide complete information may mislead readers.
Lines 69-71: A) Specifying the taxonomic keys used for tick identification is recommended.
- B) Did the authors only work with adult female ticks, or did they also work with other stages (e.g., nymphs)? The methodology does not clarify this.
Line 72: It is “°C”. Please correct this. It is unclear.
Section 2.3: Why was an Alpha diversity analysis not performed on the samples?
Lines 103-104: Please indicate which reference sequences of NCBI were used for the analysis.
Sections 3.2., 3.3., 3.4., and 3.5: A) What is the coverage and depth of the sequences for each genus (Anaplasma, Ehrlichia, Rickettsia, Babesia, and Theileria)? It is not mentioned anywhere in the manuscript, but in lines 96-98, the authors say that sequencing was performed for all positive amplicons.
- B) How can the authors ensure their "Candidatus" records are new pathogen species? Couldn't it be sequencing or amplification errors?
Reviewer 3 Report
Comments and Suggestions for Authors
The manuscript is about diversity and coinfections of pathogens in ticks from ruminants in Pakistan.
Authors’ list: Line 5, and *?? Kindly correct.
Line 14: Ruminants are key in pathogen transmission. How??
In background, there is no detail of prevalence of ticks and tick-borne pathogens, and coinfection of pathogens from ruminants in Khyber Pakhtunkhwa Province.
Line 61-62: The collection sites' latitude, longitude, and host animals were recorded during tick collection. There is no detail in the manuscript. Authors may include this data in supplementary table.
Line 62-63: Tick collection was carried out with local authorities' support and approval, including the Civil Veterinary Hospital and Department of Animal Husbandry, Peshawar. Where is ethical approval? You cannot publish this article without ethical approval. Write ethical approval number.
Line 69: Kindly add reference of taxonomic keys used by entomologist.
In methods, there is no detail of number of samples. What is the study design? How many animals were sampled? How many ticks were collected? How many ticks were subjected to DNA extraction? There is no detail of use of positive and negative control in the PCR.
Results: A total of 989 tick species were collected from the 19 sites of three districts in Khyber Pakhtunkhwa Province, Pakistan (Figure 1), comprising 867 females and 122 males from 123 cattle, goats and sheep. Did you find any immature stage of ticks? It is very unlikely that you find only male and female stages.
Line 125: The collected ticks belonged to five species of the family Ixodes??? family Ixodidae
The sequences of A. marginale-positive samples were closely related to each other, with nucleotide identities of 99.5-100%, 97.2-100%, and 98.2-100% for the 16S rRNA, gltA, and groEL gene, respectively. For species level identification, similarity may be 98% or more. Please add similarity table in supplementary data using GenBank.
"Best Match Species, Accession Number, Sequence Identity %, Sequence Coverage %, E-value, Host, and Country"
Where are accession numbers for Anaplasma species?
Figure 2, 3, 4, and 5, the numbers at nodes of phylogenetic trees are low. Provide explanation of support of the nodes and species identity.
There is no separate conclusion.
Comments on the Quality of English LanguageKindly proofread manuscript to remove English language errors.
Round 2
Reviewer 1 Report
Comments and Suggestions for Authors
Comments to the manuscript: High diversity and low coinfections of pathogens in ticks from ruminants in Pakistan – revised version
Dear Editor and authors,
I have looked through the revised manuscript and noted that the authors in several cases have made the requested changes to the manuscript. The introduction is improved, which also applies to the M&M. The requested statistical analyses are now provided.
I do however also note that the authors have agreed to certain changes in their reply but updated the manuscript accordingly. They state that “we have revised the sentence to reflect a broader ecological interpretation without implying direct competitive dynamics”. – but the text is the same as in the first version. They also state that they have improved the readability of the figures – but to me, they look the same – except for figure 2 - that now has gone missing! Whether the discussion has been updated to the statistical assessment is debatable.
So I recommend that the authors do as they stated in the reply, and ensure that it has the requested effect on the readability of the paper. Pls., print the paper and see if you can read the figures before you resubmit.
Some minor comments.
Line 41: I suspect that you mean prevalence – not incidence.
Line 56: Delete the word “phenomena”.
Line 212: “identical to” instead of “identity from”.
Reviewer 3 Report
Comments and Suggestions for Authors
Authors complied with most of comments.
However, I have still few more comments.
In introduction, line 44-47, "(CCHFV) is transmitted by Rhipicephalus and presents a serious risk to public health in Pakistan, resulting in severe hemorrhagic fever…. (10-13)". Hyalomma ticks are considered main vectors of CCHFV. Kindly read more papers specific to CCHFV transmission. Reference 10 is about Babesia species, how it can fit in CCHFV discussion and transmission. Reference 12 is about ‘Assessing the influence of climate change and environmental factors on the top tick-borne diseases in the United States”. Please add correct references.
From response 10, add similarity table of A. marginale in supplementary data.
Best of luck!
